# Unselective Measurement of Tumor-to-Stroma Proportion in Colon Cancer at the Invasion Front—An Elusive Prognostic Factor: Original Patient Data and Review of the Literature

**DOI:** 10.3390/diagnostics14080836

**Published:** 2024-04-18

**Authors:** Zsolt Fekete, Patricia Ignat, Amelia Cristina Resiga, Nicolae Todor, Alina-Simona Muntean, Liliana Resiga, Sebastian Curcean, Gabriel Lazar, Alexandra Gherman, Dan Eniu

**Affiliations:** 1Department of Oncology, “Iuliu Hațieganu” University of Medicine and Pharmacy, 400006 Cluj-Napoca, Romania; suteu.patricia@umfcluj.ro (P.I.); sebastian.curcean@gmail.com (S.C.); gabrielllazar@yahoo.com (G.L.); gherman.alexandra@umfcluj.ro (A.G.); tudor.eniu@umfcluj.ro (D.E.); 2“Prof. Dr. I. Chiricuță” Oncology Institute, 400015 Cluj-Napoca, Romania; todor@iocn.ro (N.T.); muntean.alina@yahoo.fr (A.-S.M.); liliana_resiga2002@yahoo.com (L.R.); 3Nicolae Stăncioiu Heart Institute, 400001 Cluj-Napoca, Romania; amelia.resiga@yahoo.com

**Keywords:** fibrosis, tumor stroma, stroma-to-tumor ratio, pathology markers, colon cancer, adjuvant treatment

## Abstract

The tumor-to-stroma ratio is a highly debated prognostic factor in the management of several solid tumors and there is no universal agreement on its practicality. In our study, we proposed confirming or dismissing the hypothesis that a simple measurement of stroma quantity is an easy-to-use and strong prognostic tool. We have included 74 consecutive patients with colorectal cancer who underwent primary curative abdominal surgery. The tumors have been grouped into stroma-poor (stroma < 10%), medium-stroma (between 10 and 50%) and stroma-rich (over 50%). The proportion of tumor stroma ranged from 5% to 70% with a median of 25%. Very few, only 6.8% of patients, had stroma-rich tumors, 4% had stroma-poor tumors and 89.2% had tumors with a medium quantity of stroma. The proportion of stroma, at any cut-off, had no statistically significant influence on the disease-specific survival. This can be explained by the low proportion of stroma-rich tumors in our patient group and the inverse correlation between stroma proportion and tumor grade. The real-life proportion of stroma-rich tumors and the complex nature of the stroma–tumor interaction has to be further elucidated.

## 1. Introduction

Any solid cancer has two components: tumor cells and stroma. Neoplastic cell groups, without the capacity to stimulate vessel and stroma growth, will remain subclinical.

The tumor stroma consists of a matrix (basement membrane and extracellular matrix), cells (fibroblasts, immune cells) and vasculature. The tumor stroma is a modified connective tissue, which might promote growth, invasion and metastasis. Carcinoma-associated fibroblasts (CAFs) constitute a major portion of the reactive tumor stroma and play a central role in tumor progression. This transformation is driven by cytokines secreted by tumor cells [1].

Colorectal cancer is the third most common type of cancer in men and the second in women [2]. Most patients, in the early stages, have an excellent prognosis. Still, a minority of patients have more aggressive early disease. We cannot satisfactorily predict which patients are at higher risk of relapse and thus need more aggressive multimodal treatment. There is some evidence that the area covered by stroma on a microscopical cross section of the tumor might give hints about the tumor’s aggressiveness. The data regarding the different studies examining the methodology and role of the tumor-to-stroma ratio (TSR) are described in detail in the Discussion section. In general, a stromal content of more than around 50% is assessed by the different authors, although different cut-off values are also used for very high-stroma tumors. Both optical and automated readings of stroma are used [3]; TSR is not yet adopted in clinical guidelines, and we consider its adoption early. In our study, we tried to confirm or dismiss the hypothesis that a simple, unselective optical measurement of stroma quantity at the invasion front is an easy-to-use and strong prognostic tool.

## 2. Materials and Methods

We included 74 consecutive patients with colorectal cancer who underwent curative abdominal surgery as the first therapeutic step in the period between January 2006 and December 2013 at our tertiary academic cancer unit. We thought it was appropriate to merge rectal and non-rectal colon cancer, since other study groups have carried out the same. Inclusion criteria were colon adenocarcinoma treated curatively in the above period and a standardized pathology examination. Exclusion criteria were synchronous colon or other cancer types, neoadjuvant treatment, positive resection margins, very early colon cancer (pT1/2 N0M0, since it has very good prognosis), incomplete follow-up and low-quality pathology specimens.

Neoadjuvant treatment was excluded as it can increase the percentage of stroma by tumor cell apoptosis [4]. We also excluded mucinous tumors, known to have a worse prognosis.

We chose to evaluate surgical specimens instead of biopsies in order to have the whole tumor analyzed. We selected representative regions from the tumor.

In all our cases, one pathology specialist with more than 20 years of experience selected the most invasive part of the resected tumor (the portion which is used to establish the T denominator of the tumor and comprises “the invasion front”, defined as the region with the deepest infiltration) [5]. From this portion, 5 µm thick sections were acquired, colored with hematoxylin–eosin (HE) staining and examined under conventional microscopy.

The invasion front was pinpointed, searching with 2.5× or 5× objectives. This region was then examined through a 10× objective and the selection of an area with both tumor and stromal cells was performed. For the field of view (FOV), an area with tumor cells on all margins was chosen. When mucus was present, the pathologists visually excluded it from the area.

The percentage of stroma was estimated by two examiners (L.R. and A.R.). The tumors were grouped into three categories, stroma-poor (stroma < 10%), medium-stroma (between 10 and 50%) and stroma-rich (over 50%), based on earlier publications [6,7,8]. For representative microscopic sections, see Figure 1, Figure 2 and Figure 3.

We included in our study all classical prognostic factors, such as gender, stage, histological subtype, grading, lymphovascular and perineural invasion, tumor size and location, treatment type and sequence. We also included in our analysis body mass index, the presence of diabetes, the individual surgical team, distance from the anal verge and presentation in complete occlusion. All patients underwent an R0 resection. Patients were treated according to local guidelines based on current ESMO and NCCN recommendations. The microsatellite status was not evaluated at the time of the analysis.

The statistical analysis has been performed in Excel Microsoft Office 2007. With the chi-square test for distinct values (with Yate’s correction), we analyzed the risk factors for relapse. Survival analysis was performed with the help of Kaplan–Meier curves and log-rank test. A multivariate analysis was performed with Cox regression. A *p* value of *p* ≤ 0.05 was considered statistically significant.

Staging was performed according to 7th edition of the American Joint Committee on Cancer Staging (AJCC) Manual (2010).

## 3. Results

The ages of the patients ranged from 34 to 79, with a median of 59. Of the 74 patients, 22 (29.7%) had non-rectal colon cancer and 52 (70.3%) had rectal cancer; 38 (51.40%) were males. Patient characteristics can be observed in Table 1.

Of the 60 rectal cancer patients, 36 underwent anterior resection with sphincter sparing, 1 patient underwent anterior resection with temporary stoma and 23 underwent abdomino-perineal resection. Of the 14 colon cancer patients, 8 patients underwent limited segmental resection and 6 patients underwent hemicolectomy. Out of the 60 rectal cancer patients, 44 required adjuvant chemoradiation. The tumor size of the pathologic specimen measured between 1.5 and 10 cm, with a median of 5 cm.

The median follow-up was 75.8 months (range 6.2–98.7). In this period, there were 15 deaths (20%), 12 (16%) due to cancer relapse and 3 (4%) from other causes.

There was a statistically significant difference in DSS for patients at the earlier stages than patients at the later stages of colon cancer (Figure 4).

Sex and age did not influence DSS in our patient group.

The proportion of tumor stroma ranged from 5% to 70%, with a median of 25%.

A total of 5 patients (6.8%) had stroma-rich tumors (over 50% stroma), 3 patients (4%) had stroma-poor tumors (less than 10% stroma) and 66 patients (89.2%) had a medium quantity of stroma.

We tried to define a cut-off value for the stroma proportion which could have statistical significance for the DSS, but we could not find such a value (Figure 5).

Since the median percentage of stroma was 25% for our patients, for illustration purposes, we drew the DSS for a cut-off of 25% in Figure 6.

We separately analyzed the influence of tumor stroma on local control rate and metastases, but there was no statistically significant difference. Rectal cancers were no different from other colon cancers in terms of stroma proportion.

We have to note that there was a tendency towards higher DSS for G1 and G1 tumors compared to G3 tumors (Figure 7) and there was also a tendency for the distribution of tumors with more stroma in the G1 and G1 groups compared to the G3 group (Figure 8).

A multivariate analysis with Cox regression did not yield any significant prognostic factor.

## 4. Discussion

We think that the current analysis, although performed on a lower number of patients, is important, since it could not confirm the results of the first teams that described how unselective TSR measurement is a prognostic factor in colorectal cancer, namely Huijbers et al. [6], West et al. [7] and Park et al. [8]. These differences could be explained by several factors. At first look, we could easily conclude that simply a smaller number of patients is the main causative factor, although the explanation might be more complex, as we will demonstrate below. Certainly, we included only 74 patients, versus 710, 145 and 330. At the same time, if this pathologic feature has a measurable impact only when analyzing very large numbers of patients, it means that its importance as a prognostic factor is not substantial or that there is more to this prognostic factor than sheer unselective measurement at the tumor invasion front.

In the study of Huijbers et al. [6], the authors divided the tumors into ‘stroma-high’ (>50%) and ‘stroma-low’ (≤50%), as determined a priori, to order to have maximum discriminative power. Almost 30% of the tumors were scored as stroma-high. In the stroma-high population, the 5-year OS was lower, only 69%, versus 83.4% within the stroma-low population. For the DFS, the 5-year survival rates for stroma-high and stroma-low were 58.6% versus 77.3%. In our study, only 6.8% of patients had stroma-rich tumors, and thus this might be a reason for the negative results.

West et al. [7] set a similarly high cut-off for stroma-high tumors, i.e., >53%. Almost half of their patients had stroma-high tumors, in comparison to our group, where only 6.8% presented stroma-high cancers. Stroma proportion was an independent prognostic factor for OS in their population.

In the study published by Park et al. [8], the authors chose a similar cut-off value for stroma-high tumors (>50%) and found that 24% of tumors were stroma-high. A low TSR was associated with worse prognosis and with high T and N stages.

Micke et al. argued against using TSR as a simple prognostic tool in cancer [9]. The primary aim of their study was to provide a comprehensive and objective analysis of the tumor stroma in 16 different solid cancer types, including over 2500 patients. They used immunofluorescent staining for epithelial markers and machine learning image analysis and calculated the stroma fraction for each individual cancer case. Major differences in the median levels of stroma fraction, ranging from less than 25% in renal cell carcinoma to over 70% in pancreatobiliary-type periampullary cancer, were found, but there were also large variations in one histological subtype. For example, the median value for colorectal cancer was around 25% (identical to our study), and this varied greatly from less than 2–3% to over 90%. In colon cancer, the OS was not influenced by the TSR. A higher stroma fraction was significant only in intestinal-type periampullary cancer, with a HR of 3.59. Contrarily, in periampullary pancreatobiliary-type cancer, a higher stroma fraction was associated with improved OS and a HR of 0.56. The same positive association was observed for ER-negative breast cancer, HR 0.41.

The main characteristics of the studies involving the TSR in colon cancer can be seen in Table 2.

Rani et al. showed opposite findings, namely that, in head and neck cancer, stroma-low tumors have a poorer prognosis [10].

We used only one FOV for establishing the TSR and did not use several fields to calculate an average value, as did Park et al. [8]. Micke et al. [9], who argue against the importance of the TSR, used the whole available specimen and did not select the invasion front as we did in our analysis, or Park et al. did.

In 2018, the Huijbers–Pelt study team published a guideline to rating the tumor-to-stroma ratio [11]. They described their method in more detail, stating that the region with the highest stroma proportion on the slide of the most invasive part should be decisive. Thus, they direct their measurement toward the stroma-rich area. Their method has been cited by 70 scientific articles and 8 of these are original studies which tried to replicate their findings. From these 8 studies, 5 produced positive results [12,13,14,15,16], 2 negative [17,18] and 1 quite the opposite in terms of prognostic value [19]. We think that these mixed results and operator and method dependency determined the low enthusiasm in adopting TSR measurements in clinical guidelines.

The results published by Martin et al. were the most surprising [19]. The authors showed that a high tumor proportion/low stromal proportion (≥54% of tumor cells) at the invasion front was associated with distant metastasis and worse overall survival. Additionally, a multivariate Cox regression showed that a high tumor proportion was an independent risk factor from T stage, microsatellite stability and tumor budding grade. The hazard ratio of patients with high tumor percentage was 3.2. The authors themselves called the findings “quite surprising, because, so far, only a low TP/high SP has been linked to a worse prognosis”. On the other hand, a very high stroma percentage of more than 85% was also associated with a worse overall survival. Our own findings can be easily explained in the light of these mixed results: we had an insufficient number of very high-stroma patients, which seems to bear a worse prognosis, and push survival data on the worse prognosis domain for high-stroma tumors.

Moreover, Martin et al. [18] introduced a better prognostic feature than the TSR, according to their recent publication. They introduced the term SARIFA, Stroma AReactive Invasion Front Areas. The definition of SARIFA is the “direct contact between a tumor gland/tumor cell cluster (≥5 cells) and surrounding adipose tissue in the invasion front”. The TSR was not a prognostic factor in this study.

Strous et al. [12], Aboelnasr et al. [13], Smit et al. [14], Schiele et al. [15] and Kang et al. [16] published positive results. Schiele et al. included only pT3 and pT4 tumors. All authors included mucinous tumors, which were excluded by our group.

Negative results were reported by the Dutch T1 CRC Working Group [17]. Dang et al. showed that stroma proportion is non-prognostic in T1 colon cancers, but they have not excluded that, in larger tumors, it might have a prognostic value.

Zhao et al. [20] published the largest single-cohort study yet with 814 patients. They employed both human and automated readings of stroma. A cut-off of 48.8% of stroma tissue was found to be significant. Stroma-high status was associated with reduced OS in both the discovery (HR 1.72, 95% CI 1.24–2.37, *p* = 0.001) and validation cohort (2.08, 1.26–3.42, 0.004). Their method to evaluate stroma involves the whole specimen and it is difficult to reproduce.

Eriksen et al. [21] found that stroma-rich stage II colon cancer has a worse survival and is associated with a higher rate of microsatellite stability.

**Table 2 diagnostics-14-00836-t002:** Comparison with other study teams.

Study Team/Author	Negative/Positive Study or Opposite Results	Number of Patients	Comments
The current study, Fekete et al.	Negative	74	-cut-off for stroma was 50% and 10%, but analysis was performed at any cut-off -visual analysis at invasion front, one FOV
Huijbers et al. [6]	Positive	710	-cut-off for stroma was 50% -visual analysis at invasion front, one FOV
West et al. [7]	Positive	145	-cut-off for stroma was 53% -visual analysis at invasion front, one FOV
Park et al. [8]	Positive	330	-cut-off for stroma was 50% -visual analysis at invasion front, multiple FOV
Micke et al. [9]	Negative	351	-analysis performed at any cut-off -machine learning image analysis of whole specimen
Dang et al. [17]	Negative	261	-only stage pT1, non-pedunculated tumors
Martin et al. (1) [19]	Positive and opposite	206	-higher than 53% stroma percentage associated with better survival, whereas very high stroma percentage is associated with lower survival
Martin et al. (2) [18]	Negative	445	-cut-off for stroma was 50% -visual analysis at invasion front
Strous et al. [12]	Positive	201	-cut-off for stroma was 50% -visual analysis at invasion front
Aboelnasr et al. [13]	Positive	103	-cut-off for stroma was 50% -visual analysis at invasion front
Smit et al. [14]	Positive	246	-cut-off for stroma was 50% -visual analysis at invasion front
Schiele et al. [15]	Positive	291	-cut-off for stroma was 50% -visual and automated analysis -only pT3 and pT4 tumors
Kang et al. [16]	Positive	266	-cut-off for stroma was 50% -visual analysis at invasion front -included mucinous tumors
Zhao et al. [20]	Positive	814	-cut-off for stroma was 48.8% -both human and automated measurements
Eriksen et al. [21]	Positive	573	-cut-off for stroma was 50% and 75% -visual analysis at invasion front

Rather than the TSR, the function of the stroma, and thus the proteins expressed [22] and the gene signature of the tumor–stroma complex, seems to be more important in a wider number of colon tumors [23]. For example, in breast cancer, the smooth muscle actin (SMA) expression on stromal cells (peritumoral myofibroblasts) was correlated with a higher histological grade. There was even a trend for reduced PFS in the group of node-negative tumors with strong SMA stromal expression [22]. In bladder cancer, certain IHC features of the stroma can classify tumors by aggressiveness [24].

The DNA or mRNA fragments obtained from the tumor without specific microdissection can be both tumor cell- and fibroblast-derived [25]. Isella et al., with our collaboration [23], described in rectal cancer three main subtypes according to the mARN profiles of the tumor–stroma complex: transit-amplifying/enterocyte, goblet/inflammatory and stem/serrated/mesenchymal (SSM). The SSM subtype has been more resistant to chemoradiation. The mARN profile of the SSM subtype has been dominated by mARNs released by stroma fibroblasts. These findings constitute indirect proof that amplified stroma function is a negative prognostic factor.

Another reason which might have cancelled out the importance of stroma abundance in our study is tumor grade, since tumor grade is inversely correlated with the quantity of stroma (G1 and G2 tumors have more stroma than G3 tumors), thus canceling out each other’s influence. We propose that, if one is to analyze the contribution of stroma, one should only do so after matching tumors with the same grade. This was not possible in our study since we would have needed more patients.

Lymphatic microvessel density (LMVD) is another highly debated prognostic factor with conflicting results and that is difficult to reproduce [26].

Recently, Benias et al. [27] and Cenaj et al. [28] described in more detail the human interstitium, which merits the denomination of that of a previously unrecognized organ, which is part of the submucosa and is a fluid-filled space, draining into the lymph vessels and lymph nodes. It is supported by a complex network of thick collagen bundles. These collagen fibers are intermittently lined by fibroblast-like cells that stain with endothelial markers and vimentin. There is a body-wide network of these fluid-filled interstitial spaces. This new information refines classical knowledge about the simplified local, lymphatic, hematogenous and intracavitary spread. The variable characteristics of the interstitium need to be further investigated in relation to the metastatic capability of different tumors.

Liu et al. [29], in a preprint article based on all previous data on the TSR, tried to define different subtypes of stroma. The study team showed that tumors with disorganized or heterogeneous stroma tumors with neovascularization were associated with poor survival, whereas those with aligned and organized tumor stromal “strands” were frequently among the top survival-favorable groups. There is a need to further subdivide colon cancer by microscopic subtype.

In conclusion, we reiterate that the limitations of our smaller sample size need to be addressed in future research.

## 5. Conclusions

We can conclude that a simple measurement of the TSR is not a robust and easily reproducible prognostic factor in colon cancer patients who have undergone primary surgery. The percentage of stroma-high tumors, excluding mucinous tumors, was low in our patient group. Moreover, tumor type, stroma type, T stage, tumor front characteristics, such as SARIFA, and other factors might play important roles in TSR measurement. Further international collaboration and the further adoption of automated techniques and machine-learning are needed to standardize tumor categorization and stroma measurement. The complex nature of stroma–tumor interactions has to be further explained.

## Figures and Tables

**Figure 1 diagnostics-14-00836-f001:**
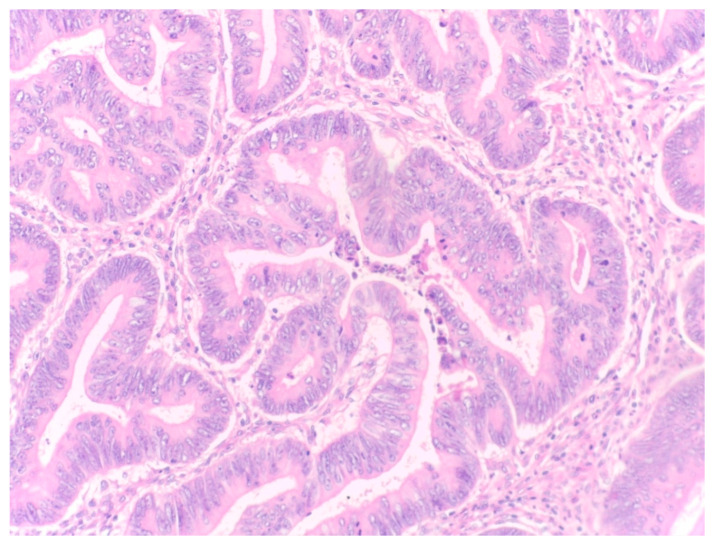
Example of a stroma-poor tumor, eosin–hematoxylin (EH) stain, magnification 200×, stroma around 5%.

**Figure 2 diagnostics-14-00836-f002:**
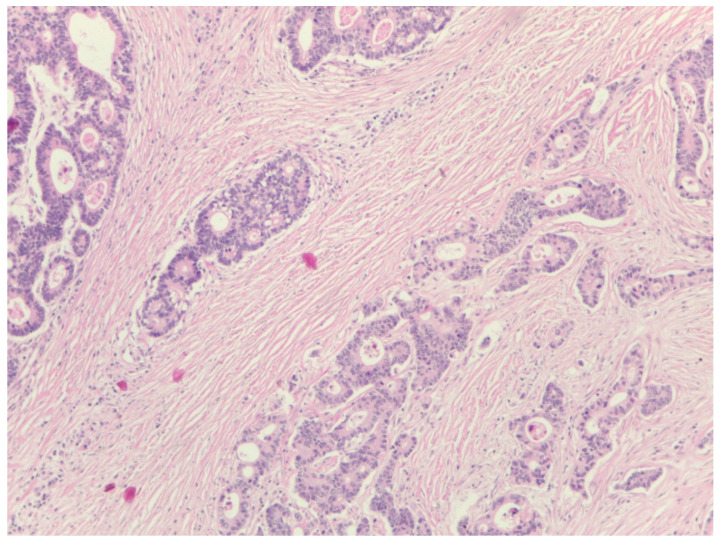
Example of a tumor with medium-content stroma, eosin–hematoxylin (EH) stain, magnification 200×, stroma around 40%.

**Figure 3 diagnostics-14-00836-f003:**
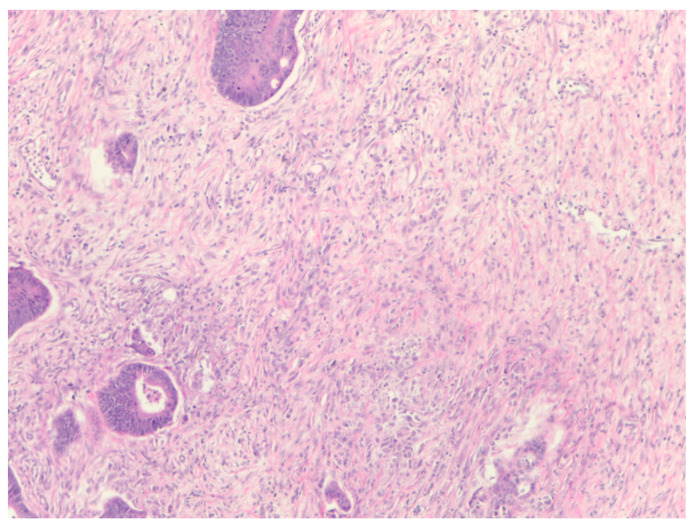
Example of a stroma-rich tumor, eosin–hematoxylin (EH) stain, magnification 200×, stroma around 70%.

**Figure 4 diagnostics-14-00836-f004:**
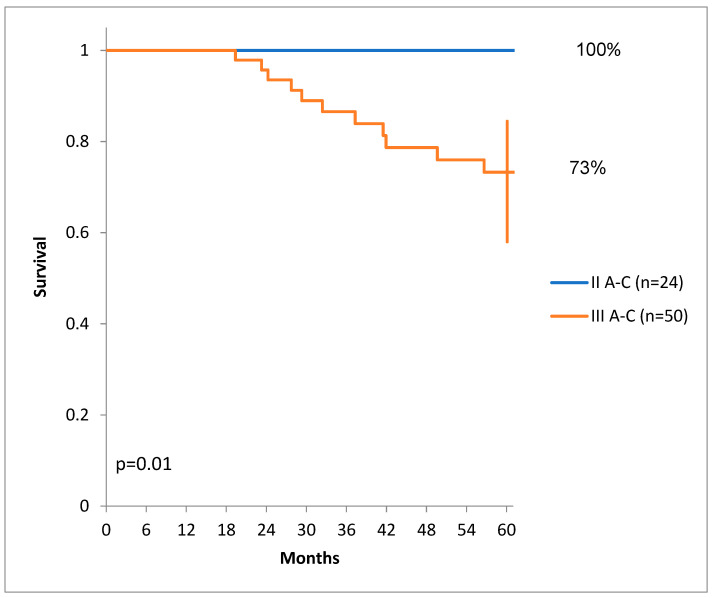
Disease-specific survival for different stages.

**Figure 5 diagnostics-14-00836-f005:**
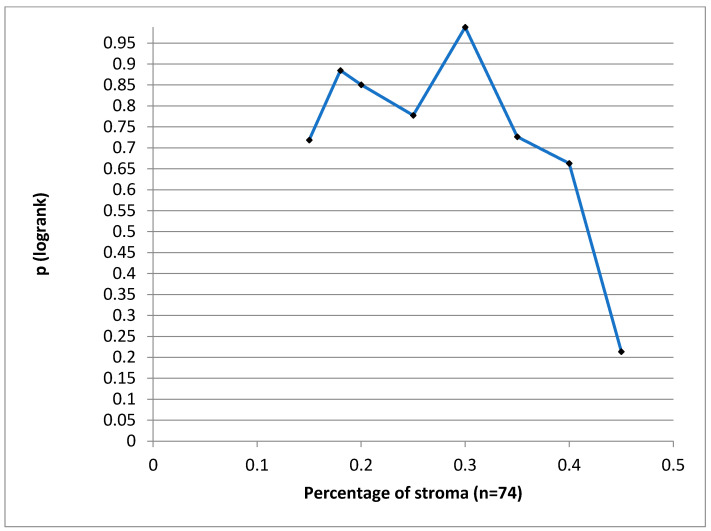
*p*-value for different cut-off values of stroma proportion.

**Figure 6 diagnostics-14-00836-f006:**
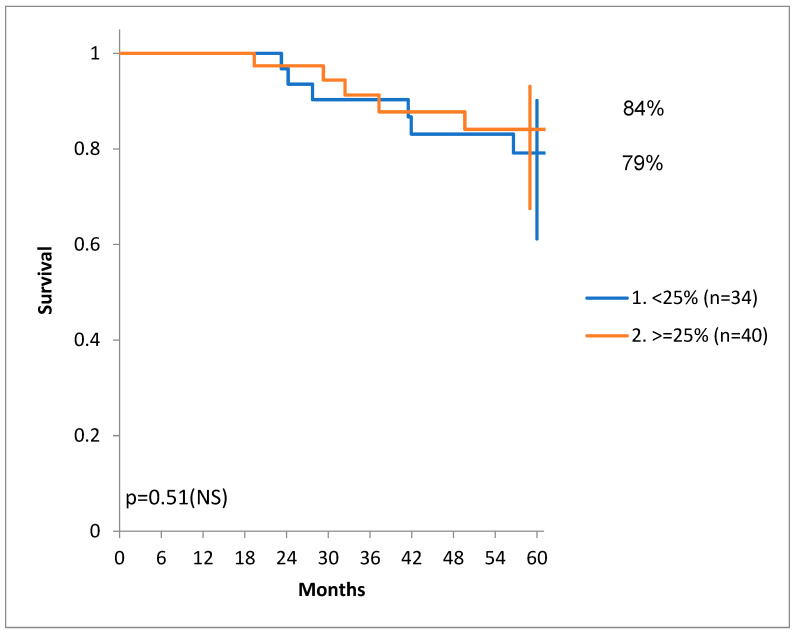
Disease-specific survival for patients with a stroma proportion less than 25% and more than 25%.

**Figure 7 diagnostics-14-00836-f007:**
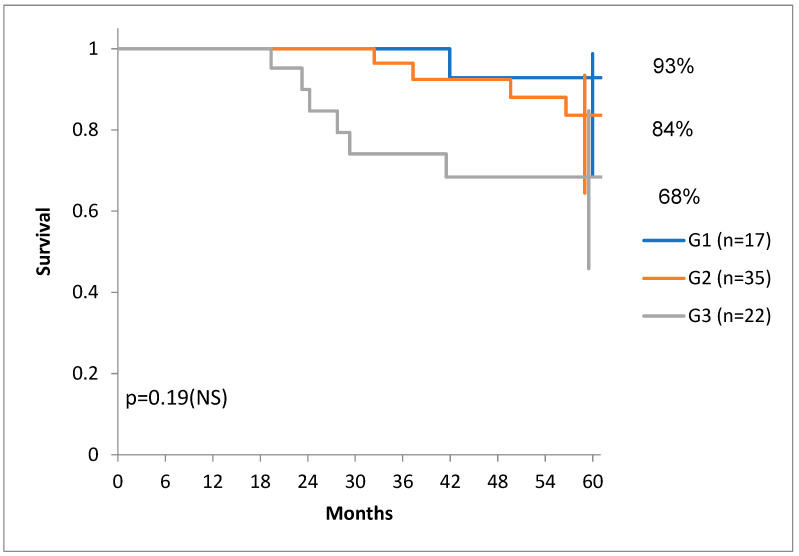
Disease-specific survival for patients with G1, G2 and G3 tumors.

**Figure 8 diagnostics-14-00836-f008:**
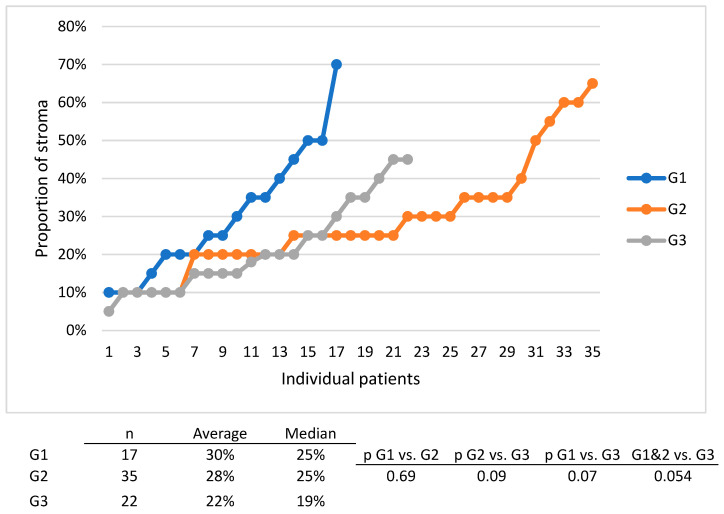
Individual patient data on the percentage of stroma for patients with G1, G2 and G3 tumors.

**Table 1 diagnostics-14-00836-t001:** Patient characteristics.

Variable	Classifier	Number	(%)
Stage (pTNM, surgical)	IIA	19	25.7
	IIB	3	4.1
	IIC	2	2.7
	IIIA	4	5.4
	IIIB	36	48.6
	IIIC	10	13.5
Subsite	Rectal	60	81.1
	Non-rectal	14	18.9
Adjuvant chemotherapy	Yes	58	78.4
	No	16	21.6
Adjuvant radiotherapy	Yes	44	59.5
	No	30	40.5
Grading	G1	17	23
	G2	35	47
	G3	22	30
Lymphatic invasion	L0	36	48.6
	L1	38	51.4
Vascular invasion	V0	69	93.2
	V1	5	6.8
Perineural invasion	pNi0	58	52.4
	pNi+	16	21.6

## Data Availability

The data presented in this study are available on request from the corresponding author. The data are not publicly available due to patient privacy.

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
