# Peer review of "Unselective Measurement of Tumor-to-Stroma Proportion in Colon Cancer at the Invasion Front—An Elusive Prognostic Factor: Original Patient Data and Review of the Literature"

_diagnostics, 2024, doi:10.3390/diagnostics14080836_

Round 1

Reviewer 1 Report (Previous Reviewer 1)

Comments and Suggestions for Authors

The previous version of this manuscript was reviewed by me, and it seems that authors improved manuscript in as much as possible. However, the authors should be mentioned low sample size as a limitation at the end of discussion.

Author Response

We appreciate your relevant feedback and assistance! The manuscript has been revised as recommended.

Reviewer 2 Report (Previous Reviewer 2)

Comments and Suggestions for Authors

Comments and concerns addressed

Author Response

Thank you again, it has been very helpful! We appreciate your input!

This manuscript is a resubmission of an earlier submission. The following is a list of the peer review reports and author responses from that submission.

Round 1

Reviewer 1 Report

Comments and Suggestions for Authors

1- sample size is low

2- Authors  must be used cox regression for cotrol of covariate

Reviewer 2 Report

Comments and Suggestions for Authors

This paper is a short communication describing an attempt to correlate tumor stroma percentage on pathological slides to patient outcomes.  Ultimately, the results were negative, but such a finding is important to share.

Despite this paper meant to a short communication, I have some suggestions:

1. In the discussion, please provide a table to summarize a comparison of your findings with those discussed in the beginning of the discussion.

2. The introduction is very short. Please summarize what is known about the colon cancer tumor stroma and patient outcomes, and what additional value your study adds, especially since there have been others published looking at similar items.

3. Can you provide any information on tumor microsatillite status, PNI, LVI, etc?

4. Is the stage and grading as per AJCC, and if so, which version?

5. Details on chemotherapy/radiotherapy regimens.  It appears that this study is primarily rectal tumors.  Is it appropriate to merge colon and rectal tumor into the same analysis.
